# Canopy Opening and Site Preparation Effects on Conifer and Understory Establishment and Growth after an Uneven-Aged Free Selection Regeneration Harvest in the Northern Rocky Mountains, USA

**Theresa B. Jain \*, Russell T. Graham, John C. Byrne and Benjamin C. Bright**

United States Department of Agriculture, Forest Service, Rocky Mountain Research Station, Moscow, ID 83843, USA; russell.t.graham@usda.gov (R.T.G.); john.c.byrne@usda.gov (J.C.B.); benjamin.c.bright@usda.gov (B.C.B.)
\* Correspondence: terrie.jain@usda.gov; Tel.: +1-208-883-2331

**Abstract:** *Research Highlights:* Forest management is trending toward creating multi-aged forest structures and diverse vegetative compositions. The challenge is successfully designing and implementing treatments that create these diverse forests. Regeneration establishment is the most important step when applying a silvicultural system because it determines future treatments and optimizes management options. This study provided the minimum canopy openings that favor the establishment of shade-tolerant and shade-intolerant tree species to inform the implementation of uneven-aged management. *Background and Objectives*: A replicated study was implemented in 2007 in moist mixed-conifer forests to design, apply, and test two silvicultural concepts, canopy opening size and site preparation. Our objective in 2015 was to evaluate tree regeneration establishment and growth and understory vegetation in relation to these two silvicultural concepts. *Materials and Methods*: Canopy opening sizes as measured by lidar ranged from 15% to 100%; and through the application of prescribed fire, mastication, pile and burn, or no site preparation, different combinations of forest floor substrates were created. We stratified our study area into five canopy opening classes and four site preparation treatments. Using this stratified sampling scheme, we located 65 plots and measured tree species, abundance, 5-year height growth, and vegetative lifeforms. *Results:* The pile and burn site preparation favored the establishment of all six tree species. The canopy opening size of 55% to 92% favored the regeneration of both shade-tolerant and shade-intolerant species. Grand-fir 5-year height growth was significantly influenced by site preparation and canopy opening, and western white pine 5-year height growth was only influenced by canopy opening. Treatments did not influence vegetative richness. *Conclusions:* This study provided key treatment parameters in designing the regeneration step for uneven-aged management strategies with the goal of creating vegetative diversity and establishing shade-intolerant tree species in moist mixed-conifer forests.

**Keywords:** mixed moist conifer; shade-intolerant tree species; western white pine; restoration; variable density thinning; multi-aged forests

---

## 1. Introduction

There is an infinite number of ways to implement forest treatments and design silvicultural systems to promote multi-aged and irregular compositions and structures over time and space; the challenge lies in the design and implementation of treatments. There are two general approaches. The first approach is to develop methods that mimic historical forest conditions (previous to the 1900s). In ponderosa pine forests, researchers have mapped and quantified the structure of remnant historical forests to develop marking guide parameters that are used to reestablish these conditions [1,2]. A good example of this

approach is called individuals, clumps, and openings [3]. This method provides specific parameters that guide the implementation of treatments and the user decides group size, spacing, and number of gaps. This method provides a formula to aid in marking trees and managers do not need special silvicultural knowledge to implement. The second approach is to incorporate forest dynamics, species silvics, gap dynamics, and other ecological concepts, in combination with the management objectives, harvest operations, and feasibility, to design the treatments [4,5]. This study uses this second approach to guide treatment designs.

Applying a silvicultural system makes each treatment unique because outcomes are dependent on the management objectives, the ecology of the forest where treatments occur, current conditions, and desired future conditions. This approach uses the silvicultural system as a template to guide treatments through time and incorporate the different phases of stand development [5]. Although this initially appears straightforward, a well-designed silvicultural system contains an artistic component because the silviculturist has to innovatively apply operational realities, critical science concepts within the context of their understanding of ecological relationships, and the landowner's management objectives, both in the short- and long-term. This approach requires greater knowledge of forest ecology, including species silvics (trees and other understory vegetation), forest and disturbance dynamics, soils, gap ecology, and variation in physical setting. This approach also requires some sophistication when designing and implementing treatments within a silvicultural system.

Free selection uses a silvicultural system approach and was formulated to introduce multi-aged, irregular forest structures and compositions in mixed-conifer forests [6]. The free selection silvicultural system does not contain discrete stand and entry metrics that define the desired future forest; rather, the system is designed to adjust stand entries using stand structure and composition, determinants which are described in the silvicultural system. These determinants depend on the objectives, but typically this system is designed to maintain high tree vigor to enhance disturbance resilience, favors disturbance-resistant species, introduces a high fuel heterogeneity, and promotes elements that enhance wildlife habitat (mature trees, snags, woody debris, and edge widths). Similar to Churchill et al. (2013), free selection also contains gaps, individual trees, and clumps of trees but its foundation is based on creating and developing a diversity of growing environments (that are sometimes referred to as "operational environments" [7]) rather than creating a particular forest structure. Opening characteristics (size, shape, and juxtaposition) and soil substrates (blackened, mineral, and organic) can shape the operational environment defined by light, heat, moisture, nutrients, root biomass, and seed availability. In the northern Rocky Mountains, the different opening sizes and soil substrates, when combined with the physical setting, influence the regeneration, vegetation composition, competition, and tree growth [8,9]. This system also allows for other disturbances (wind, disease, insects, and wildfire) to achieve target conditions. Finally, the free selection system is not a simple stand-level treatment; this system applies a multi-spatial scale approach by tending toward creating "treatment" mosaics of compositions and structures at the landscape—thus the size, shape, and juxtaposition of treatments are determined by the physical environment (steepness, benches, ridges, and riparian areas), harvest and site preparation operations, and current condition.

A replicated study was implemented at Priest River Experimental Forest in the northern Rocky Mountains, USA, to design, apply, and test two silvicultural concepts, opening size and site preparation, which would inform the subtle nuances associated with the regeneration step for the free selection silvicultural system (Figure 1). Jain et al. (2008) reported implementation feasibility using environmental assessments and typical contract language for harvesting, site preparation, and planting. Our objective is to quantify and evaluate tree regeneration establishment, tree growth, and diversity of vegetative lifeforms (trees, shrubs, forbs, ferns, and grasses) and tree species as a function of opening size and site preparation, 8 years after treatments were applied.

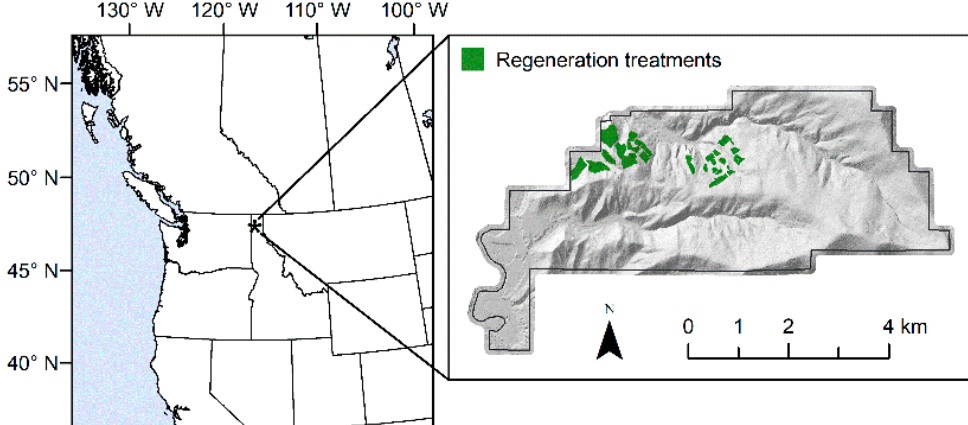

**Figure 1.** Priest River Experimental Forest (PREF) in northern Idaho, USA, and location of the Free Selection Silviculture study (green blocks) within PREF.

## 2. Materials and Methods

### 2.1. Study Area

The study is located on the Priest River Experimental Forest (PREF) in northern Idaho (Figure 1). PREF is on the west slope of the Selkirk Mountains. The Canyon Creek drainage within PREF contains the treated areas used for this study. Canyon Creek drains from the east to the west into Priest River and is therefore dominated by north- and south-facing slopes, though small side drainages do create some east- and west-facing aspects. The elevational range of the plots used for this study is 730 to 1100 m.

The climate of PREF is typically characterized by warm, dry summers and moist, moderately cold winters. This climate pattern encourages the growth of a predominately moist mixed-conifer forest, with major species being western hemlock (*Tsuga heterophylla* (Raf.) Sarg.), western redcedar (*Thuja plicata* Donn ex D. Don), Douglas-fir (*Pseudotsuga menziesii* (Mirb.) Franco), grand fir (*Abies grandis* (Douglas ex D. Don) Lindl.), western white pine (*Pinus monticola* Douglas ex D. Don), western larch (*Larix occidentalis* Nutt.), and ponderosa pine (*Pinus ponderosa* Lawson & C. Lawson). In 2007, the free selection system was implemented in the Canyon Creek drainage, by using several site preparation treatments and cutting intensities to create a variety of forest floor and canopy openings [10] (See Appendix A on free selection implementation concepts applied in 2007).

### 2.2. Experimental Design

This study uses different opening sizes and site preparation techniques to evaluate the variation in establishment and growth of regeneration by species (see Jain et al. 2008 and Table A1 for harvest and site preparation design and Figure A1 for examples of created opening sizes). Stratification of field sampling sites within the matrix of sites created within Canyon Creek for this study was guided by using geospatial data and geographic information system (GIS) techniques. Lidar data were acquired with a density of 10.4 points/m$^2$ across PREF in 2011. A grid of percent canopy cover, defined as the percentage of first returns greater than 1.37 m, was derived from this lidar data. For stratification purposes, the percent canopy cover grid was inverted and placed into five canopy opening classes (<25%, 25%–45%, 45%–55%, 55%–92%, and >92%). Using GIS software, this canopy opening class grid was overlaid on a GIS polygon layer documenting locations of site preparation treatments that were implemented in 2007 (Table A1), which included prescribed fire, pile and burn, mastication (shreds noncommercial sized trees or shrubs into small chunks [11]), and no treatment, to stratify the landscape [10].

A total of 20 strata were possible (5 canopy openings × 4 site preparation treatments) (Table 1). Existing plot locations (measured in 2004, 2005, and 2011) were overlaid on the stratification layer

and assigned the strata in which they were located. Plots visited in 2011 were given precedence for remeasurement. Additional candidate plot locations were randomly located within strata that contained no existing plots. A field crew aimed to visit at least 3 field plots within each strata, though it wasn't possible to find enough suitable sites in all strata, and 4 strata did not occur on the landscape. Canopy opening classes of 0%–25% in most treated units were particularly elusive since harvesting had reduced most stands to a greater canopy opening class. Most of the plots in the design for these 0%–25% canopy opening class/site preparation treatment strata were on the edge of the unit or completely outside of it and since these sites had residual trees, prescribed fire was not applied. This resulted in an uneven sampling, but we were able to place 65 plots across the potential 20 sampling strata.

**Table 1.** Number of plots within the canopy opening and site preparation strata.

| Canopy Opening Class (%) | No Site Preparation | Prescribed Fire | Pile and Burn | Mastication | Total |
|:---:|:---:|:---:|:---:|:---:|:---:|
| <25 | 0 | 0 | 2 | 3 | 5 |
| 25–45 | 1 | 1 | 2 | 2 | 6 |
| 45–55 | 3 | 0 | 1 | 7 | 11 |
| 55–92 | 5 | 8 | 13 | 5 | 31 |
| >92 | 0 | 5 | 2 | 5 | 12 |
| Total | 9 | 14 | 20 | 22 | 65 |

The specific opening size classes we chose were based on opening size management thresholds identified by Jain et al. (2004). The management thresholds included canopy openings <25%, which provide minimal growing space for shade-tolerant species such as western hemlock to regenerate. Canopy opening between 25%–45% facilitate regeneration establishment for moderately shade-tolerant species such as western white pine. In canopy openings of 55%–92%, shade-intolerant species such as western larch tend to regenerate and establish. For canopy openings >92%, they found that western white pine attained free-to-grow status.

*2.3. Data Collection*

Forest Floor, Trees, and Understory Vegetation

A nested plot design was used at each sampling point, with a 0.002 ha (1/200th acre) circular plot (2.5 m (8.2 ft) radius) used for measurements of substrata cover, vegetative lifeform cover, and small trees (≥0.3 m (1.0 ft) in height and <10.2 cm (4.0 inch) diameter at breast height (DBH, 1.37 m)). We took ocular estimates of the following forest floor substrata: solid wood, rotten wood, humus and litter combined, mineral soil, and blackened surface [12] using the following percent cover classes: 0%, 1%–10%, 11%–25%, 26%–50%, 51%–75%, and 76%–100%. We used the same cover classes as above to ocularly estimate the percentage cover of the following lifeforms: small trees (as defined above), shrubs, forbs/low shrubs, ferns, and grass. We counted small trees by species and height class: 0.3–1.2, 1.2–2.1, 2.1–3.0, 3.0–4.0, 4.0–4.9, 4.9–5.8, 5.8–6.7 m (1–4, 4–7, 7–10, 10–13, 13–16, 16–19, 19–22 ft). For each species/size class combination (except western redcedar and western hemlock), we measured three trees for total height and heights 1, 2, 3, 4, and 5 years back by counting branch whorls using a height pole, to the nearest 0.01 m (0.05 ft). For western redcedar and western hemlock, however, it was only possible to measure the current height and height 1 year back. On these trees, we also measured diameter at root collar (DRC), and DBH for small trees greater than 1.37 m (4.5 ft) total height, with a caliper to the nearest millimeter. The three sample trees in each species/size class were measured from a different section of the plot, if available.

A 0.04 ha (1/10th acre) circular plot (11.3 m (37.2 ft) radius) was used for measurements of large trees (≥10.2 cm DBH). For each of the larger trees in the 0.04 ha plot, we recorded a unique tree identification number, distance from plot center (to the nearest 0.03 m (1/10th ft)), azimuth from plot center to the tree (degrees), status (1 = live, 8 = dead), species, and DBH (to the nearest 0.25 cm (1/10th in)). We also recorded any relevant conditions, such as broken top for snags (dead trees) or if the tree or snag was damaged by fire. For each tree species, we measured total height and crown

base height (both to the nearest 0.03 m) on the trees with the smallest and largest DBH, as well as another tree between the smallest and largest DBH, resulting in three measurements per species (if available). Crown base height was considered the lowest live branch. We did not measure heights on snags. Some of the plots were originally measured in 2011 and trees ≥ 10.2 cm were stem-mapped on a 0.017 ha (1/24th acre) plot. We revisited these plots and we added trees that were within an 11.3 m radius that were not included before because they were farther than 7.5 m (24 ft) from plot center. We also visited plots that were established in 2004 and 2005; for these plots, we overlaid the current plot design and recorded measurements for all trees on the 0.04 ha plot.

### 2.4. Statistical Analysis

Though small trees were counted by species within seven height classes, we used the sum of the trees across all height classes within each species for analysis of small tree abundance by species as a function of site preparation treatment and canopy opening. We used a Pearson's Chi-square goodness-of-fit statistic for this analysis, with significance at alpha < 0.05, where we tested two null hypotheses: (1) frequency of trees by species are independent of site preparation and are identically distributed among the different site preparation methods, and (2) density of trees (trees ha$^{-1}$) by species are independent of canopy opening and are identically distributed among the different canopy opening size classes. We present our results as chi-grams, which use expected values of 0 and show the observed deviation either equal to the expected or more or less abundant than the expected.

$$deviation = \frac{observed - expected}{\sqrt{expected}}$$

We used mixed models regression to analyze: (1) 5-year height increment (cm) and site preparation treatment and canopy opening for western white pine, western larch, Douglas-fir and grand fir, (2) for understory vegetation cover (%), we used a mixed model analysis of variance with site preparation and canopy opening as fixed effects, (3) for substrate, which is influenced by site preparation, we conducted a mixed model analysis of variance with site preparation as the fixed effect and (4) vegetation diversity which was defined as absence or presence of the different lifeforms and tree species, the total number ranged from 5 to 12 different lifeforms and tree species. Data using percent cover, because it was a proportion, was transformed using the arcsine of the square root to meet model assumptions. We used SAS 9.4 software GLIMMIX procedure [13] for all statistical analyses and R software [14] to create the deviation graphs in the paper. For vegetation diversity we used the GLIMMIX procedure but used a negative binomial distribution. In all analyses, we tested for a site preparation and canopy opening interaction; however, all tests were not significant so we only report site preparation and canopy opening in our figures and tables.

### 3. Results

The remaining overstory trees (dead and alive), eight years after harvest, varied depending on the opening size (Table 2). Most delayed mortality (average basal area of 7.1 m$^2$ ha$^{-1}$) occurred on sites with <25% canopy openings. Basal area was similar in canopy opening classes of 25% to 45% and 45% to 55%, with average basal areas of 25.9 and 22.3 m$^2$ ha$^{-1}$, respectively, but more variation in basal area occurred in the 25% to 45% canopy opening classes. Sites that had >92% canopy opening had an average basal area of <3 m$^2$ ha$^{-1}$ with tree densities that ranged from 0 to 7.7 m$^2$ ha$^{-1}$. Jain et al. (2004) noted that basal area varied substantially across canopy opening sizes, depending on the tree species, and we found similar results.

**Table 2.** Remaining live and dead overstory tree density (basal area) and diameter (quadratic mean diameter (QMD) by canopy opening class.

| Canopy Opening (%) | Plots (N) | Live Basal Area (m² ha⁻¹) | | | | Dead Basal Area (m² ha⁻¹) | | | | QMD (cm) | | | |
|---|---|---|---|---|---|---|---|---|---|---|---|---|---|
| | | Min | Max | Mean | Std Dev | Min | Max | Mean | Std Dev | Min | Max | Mean | Std Dev |
| <25 | 5 | 20.1 | 47.7 | 35.6 | 11.8 | 0.0 | 23.1 | 7.1 | 9.5 | 31.9 | 55.4 | 40.3 | 9.1 |
| 25–45 | 6 | 3.1 | 44.5 | 25.9 | 13.6 | 0.0 | 13.1 | 2.3 | 5.3 | 17.9 | 44.6 | 35.0 | 9.4 |
| 45.1–55 | 11 | 12.6 | 45.0 | 22.3 | 9.3 | 0.0 | 12.3 | 3.5 | 4.5 | 21.2 | 46.7 | 37.0 | 8.0 |
| 55.1–92 | 31 | 0.0 | 24.4 | 10.3 | 6.2 | 0.0 | 13.6 | 2.1 | 3.9 | 0.0 | 74.7 | 37.1 | 18.4 |
| >92 | 12 | 0.0 | 7.7 | 2.9 | 2.7 | 0.0 | 14.2 | 1.8 | 4.2 | 0.0 | 63.0 | 29.1 | 24.1 |

Depending on the site preparation, a diversity of forest floor conditions was created (Figure 2). In places with no site preparation, the forest floor consisted of solid and rotten logs and humus and litter. In places with site preparation, most of the solid and rotten log cover was removed, exposing humus and litter cover (Figure 2a–c). However, with mastication, by the nature of the treatment, there was an increase in solid wood (Figure 2a). The pile and burn and prescribed fire site preparation tended to expose more mineral soil than the masticated sites (Figure 2d).

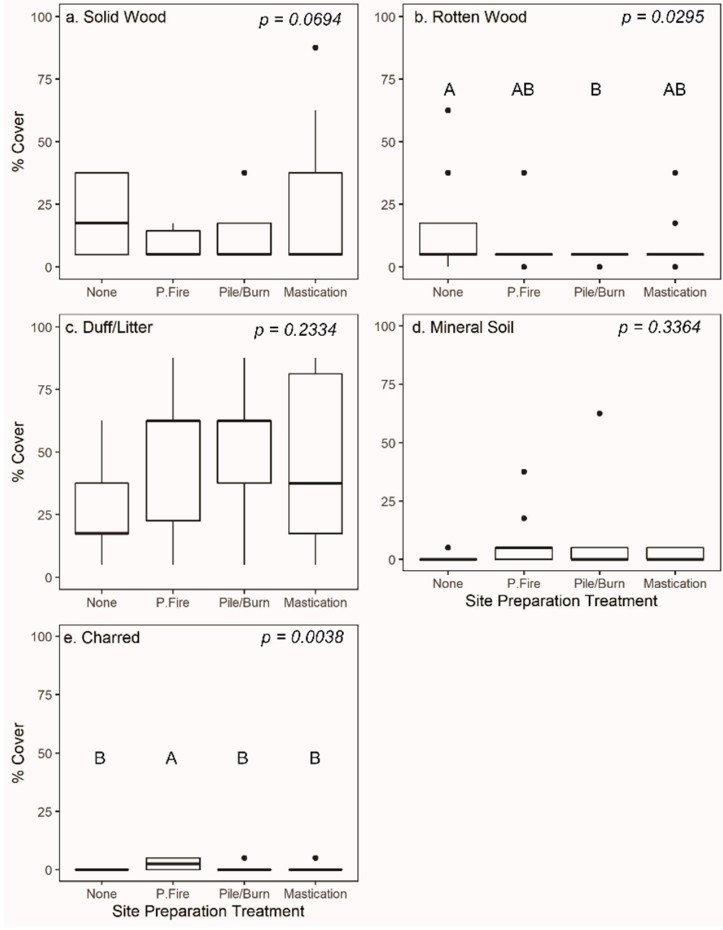

**Figure 2.** Box plots indicating the variation in forest floor components by site preparation method. Forest floor components include: solid wood which includes sticks and undecomposed wood (**a**), rotten wood includes red rotten or white rotten wood (**b**), litter (needles and leaves) and duff (humus and organic root mass) (**c**), mineral soil at the surface (**d**), and charred or blackened soil (**e**) Litter includes humus (sometimes termed duff). The amount of solid and rotten wood varied, depending on the site preparation, and more mineral soil was exposed in site preparation methods that included burning (prescribed fire and pile and burn). Letters indicate significant differences among site preparation methods.

Regeneration of tree establishment and diversity was significantly different among the different site preparation techniques and canopy opening classes identified by Jain et al. (2004) (Figure 3). Regeneration establishment was the most successful for all six species on the pile and burn site preparation method. In addition, western white pine and grand fir successfully established on the mastication site preparation (Figure 3c,d), whereas western larch (Figure 3a) successfully established on the prescribed fire site preparation. In moist forests in the northern Rocky Mountains, natural regeneration is not lacking, however, the species composition of natural regeneration differs among opening sizes. The opening size between 55% and 92% canopy opening favored successful regeneration establishment for all six species we evaluated. Although regeneration did occur in other opening sizes, they tended to be less than expected using the chi-square test (Figure 4). For example, for western hemlock there was an average of 5144 trees ha$^{-1}$ that regenerated across all the opening sizes; with most of the regeneration occurring between 25% and 92% canopy opening (Figure 4f). In contrast, western larch had an average of 526 trees ha$^{-1}$ regenerating across all canopy opening sizes with most of these trees regenerating above 55% canopy opening (73% of the total regeneration of this species) (Figure 4a). These results do not emphasize the lack of regeneration establishment, but rather the opening sizes that had the greatest diversity of species that regenerated.

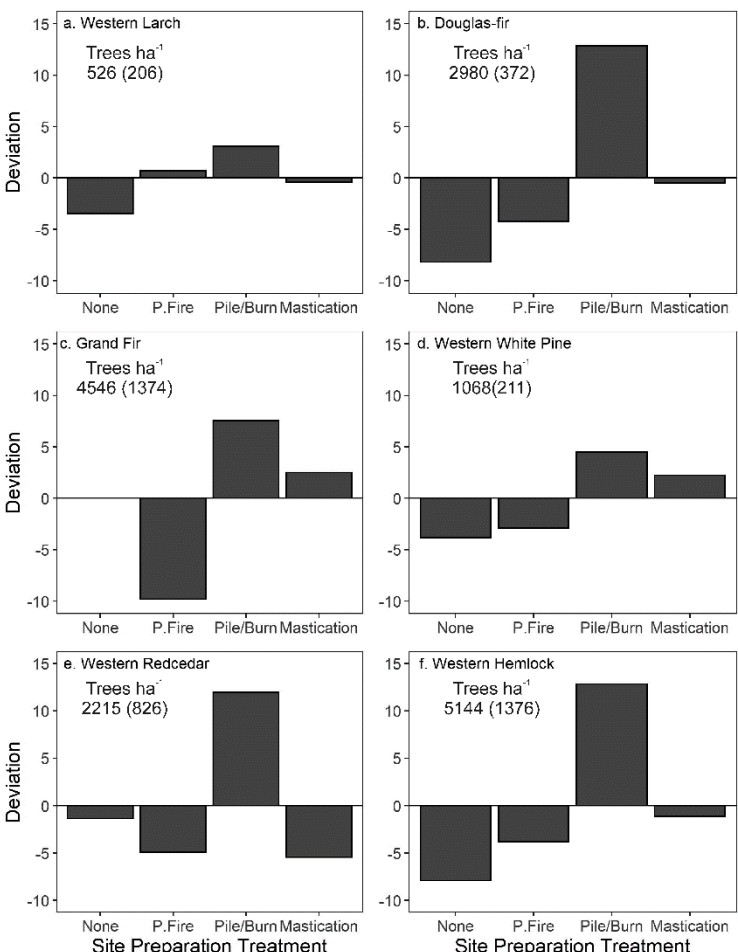

**Figure 3.** Chi-gram illustrating the difference in regeneration establishment for different site preparation methods for six conifer species. The six species are western larch (**a**), Douglas-fir (**b**), grand fir (**c**), western white pine (**d**), western redcedar (**e**) and western hemlock (**f**). Chi-grams illustrate expected versus observed. Expected values are 0 compared to the observed deviation which are either equal to the expected or more or less abundant than the expected. Chi-square goodness-of-fit was *p* = 0.0001. Note that "P.Fire" on graphs refers to Prescribed Fire. Trees ha$^{-1}$ are means and standard errors in parentheses.

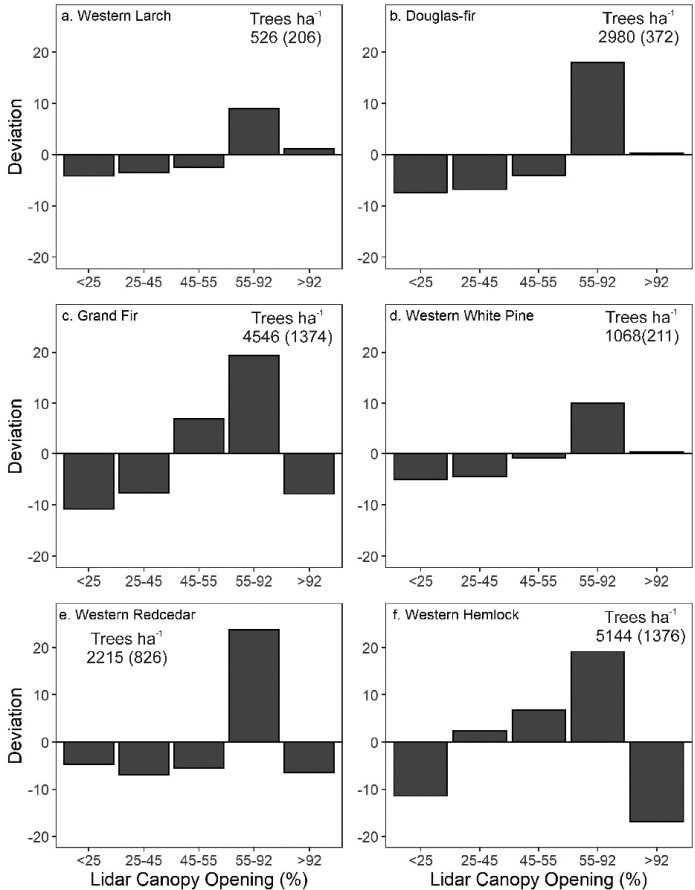

**Figure 4.** Chi-gram on regeneration establishment across a range of canopy opening sizes for six conifer species ((**a**) for western larch, (**b**) for Douglas-fir, (**c**) for grand fir, (**d**) for western white pine, (**e**) for western redcedar, and (**f**) for western hemlock) using management thresholds identified by Jain et al. (2004). Chi-grams illustrate expected versus observed. Expected values are 0 compared to the observed deviation which are either equal to the expected or more or less abundant than the expected. Chi-square goodness-of-fit was $p = 0.0001$. Trees ha$^{-1}$ are means and standard errors are in parentheses.

The influence of site preparation and canopy opening on 5-year height growth varied by species. For the least shade-tolerant western larch, site preparation ($p = 0.1492$) and canopy opening ($p = 0.9338$) did not significantly influence 5-year height growth. Similarly, the more shade-tolerant Douglas-fir 5-year height growth was not influenced by either site preparation ($p = 0.2509$) or canopy opening ($p = 0.1611$). In contrast, the moderate shade tolerant species, grand fir and western white pine, did have a significant relation in 5-year growth to either site preparation, canopy opening, or both (Figure 5). Grand-fir was significantly influenced by site preparation ($p = 0.05$) and canopy opening ($p = 0.0001$). This species had better growth on masticated, organic, and pile and burn versus on prescribed fire sites. Western white pine 5-year height growth was strongly influenced by canopy opening ($p = 0.0004$) only.

Understory vegetation, particularly shrubs and small tree cover, varied by site preparation and canopy opening (Table 3). Across the different site preparation methods, small tree cover averaged 50% on sites that were pile and burned with the lowest average cover of 19% on sites with prescribed fire. Shrub cover had high percent cover on masticated sites but also on sites that had prescribed fire and no site preparation. The pile and burn had the lowest shrub cover, most likely from the growing space being filled with young trees. When we related lifeform to canopy opening, shrubs, which tend to prefer more light, also were most abundant in opening sizes >55%. Ferns also were shown to have a significant relation with canopy opening, where they had a slight increase in cover in openings >92%. Small tree cover, which included all tree species, tended to be distributed across all

the opening sizes, but the amount of cover was not significantly different across the opening sizes, which ecologically makes sense because total small tree cover includes the full range of shade tolerant tree species. Grass and forbs were not related to either forest cover or site preparation.

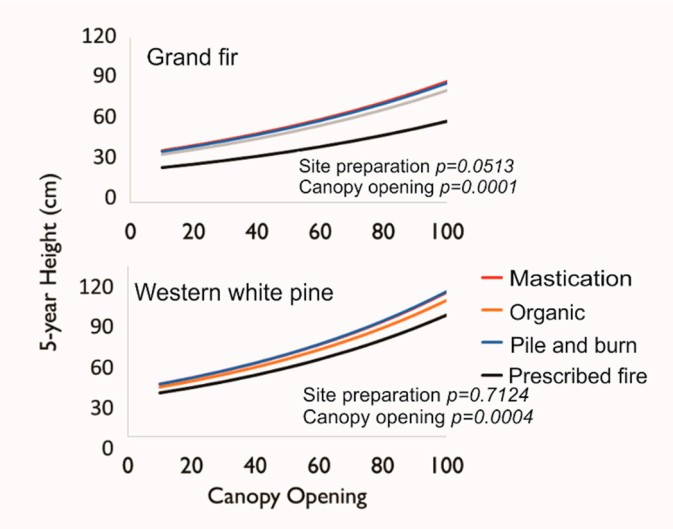

**Figure 5.** The influence of canopy opening and site preparation on 5-year height growth. Grand fir and western white pine were the only species where growth was influenced (statistically significant) by either canopy opening, site preparation, or both. Western larch (site preparation was $p = 0.1492$ and canopy opening was $p = 0.9338$) and Douglas-fir (site preparation was $p = 0.2509$ and canopy opening was $p = 0.1611$) 5-year height growth were not related to either.

**Table 3.** Understory lifeforms relation to site preparation and canopy opening classes. Percent cover for small trees, shrubs, forbs, ferns, and grass across the different site preparation methods and canopy openings. Small trees and shrubs were significantly related to site preparation. Shrubs and ferns were significantly related to opening size ($p \leq 0.05$). Means with standard errors in parentheses. Letters indicate significant differences among the means.

| Fixed Effects | N | Small Tree Cover (%) | Shrub Cover (%) | Forb Cover (%) | Fern Cover (%) | Grass Cover (%) |
|---|---|---|---|---|---|---|
| | | | Site preparation | | | |
| *p*-value | | $p = 0.0019$ | $p = 0.0066$ | $p = 0.2042$ | $p = 0.1721$ | $p = 0.3093$ |
| None | 9 | 26 (5) ab | 21 (11) ab | 26 (9) a | 1 (1) a | 4 (2) a |
| Prescribed fire | 14 | 19 (4) b | 31 (7) ab | 14 (5) a | 18 (6) a | 15 (6) a |
| Pile & burn | 20 | 50 (5) a | 17 (2) b | 24 (5) a | 5 (1) a | 12 (3) a |
| Mastication | 21 | 32 (6) ab | 28 (5) a | 13 (2) a | 7 (4) a | 9 (2) a |
| | | | Canopy opening | | | |
| *p*-value | | $p = 0.2315$ | $p = 0.0027$ | $p = 0.1164$ | $p = 0.0191$ | $p = 0.1164$ |
| <25% | 5 | 26 (12) a | 10 (0) ab | 10 (0) b | 0 (0) b | 2 (2) a |
| 25–45% | 5 | 21 (8) a | 15 (5) ab | 10 (0) ab | 14 (9) ab | 10 (0) a |
| 45.1–55% | 11 | 30 (8) a | 14 (2) b | 19 (6) b | 1 (1) b | 5 (2) a |
| 55.1–92% | 31 | 40 (4) a | 27 (4) a | 23 (4) ab | 6 (6) ab | 11 (2) a |
| >92% | 12 | 29 (6) a | 35 (7) a | 14 (4) a | 21 (8) a | 17 (6) a |

Vegetation diversity did not differ among the canopy opening or site preparations (Figure 6). Although the pile and burn site preparation and larger openings tended to have more different lifeforms and tree species, there was sufficient variation to not detect a significant difference. Although there was not a significant interaction between site preparation and canopy opening, more tree species and lifeforms tended to be reflected in the different canopy openings compared to the site preparation methods.

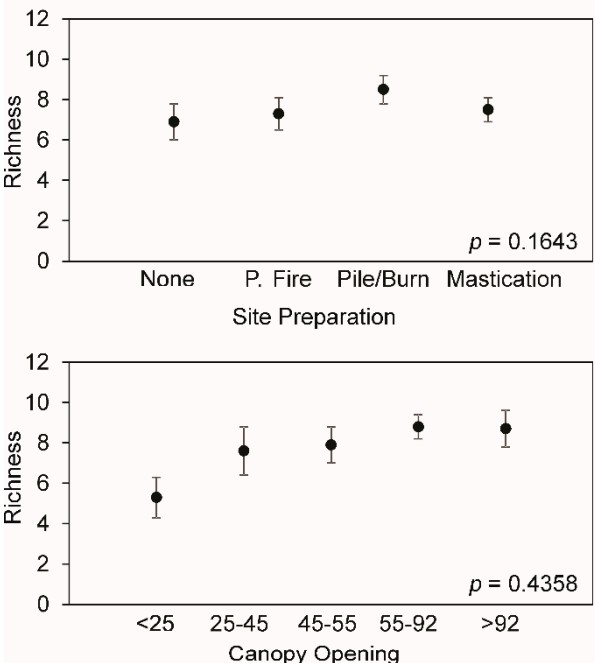

**Figure 6.** Vegetation diversity (richness) in relation to site preparation and canopy opening. Richness is the number of lifeforms and tree species present. Means and standard errors.

## 4. Discussion

This study's purpose was to evaluate the effectiveness of vegetation richness, regeneration establishment, and early growth of northern Rocky Mountain tree species in gaps and openings and different soil substrates to determine the regeneration and the most commonly applied site preparation parameters in the western United States for the free selection silviculture system. We identified that a canopy opening size from 55% to 92% favored the most diverse and abundant tree species. We also identified how site preparation can influence which species are favored based on the created forest floor and soil substrate. Although vegetation richness tended to be favored in the pile and burn and larger openings, we did not identify a significant difference.

### 4.1. Site Preparation and Regeneration

The pile and burn site preparation favored the regeneration of all the tree species we evaluated, so what made this particular site preparation method so unique? Haig et al. (1941) initially found that different northern Rocky Mountain tree species regenerate better depending on whether the regeneration bed is litter, mineral soil, or blackened surface. For example, for blackened surfaces, such as those created by prescribed fire, western larch is well suited for establishing because of its rapid root growth that can keep pace with receding soil moisture [15]. Western hemlock, because of slow root growth, tends to establish on deep organic layers or decayed logs, where moisture and nutrients are available and where there is minimum competition [16]. Western white pine germinates and establishes on mineral soil, blackened surfaces, and organic surfaces, and can establish across a wide range of light conditions [17]. The pile and burn site preparation tends to have a variety of soil substrates: mineral soil created by the machine tracks, blackened soil from the pile burning, and litter and humus in undisturbed areas (Figure 2). This diversity of organic and soil surfaces most likely favored establishment of a diversity of tree species.

Site preparation also favored different understory species abundance (Table 3). Shrubs had the highest percent cover on the prescribed fire and mastication treatments, possibly due to re-sprouting after disturbance from fire and masticating equipment. The least disturbed no site preparation sites had the highest cover of forbs, reflecting the high forb cover of undisturbed sites on these moist, mesic

sites. The pile and burn sites also had high forb cover, which may be reflective of different types of forbs, those preferring undisturbed sites and a few liking a more disturbed substrate. The opposite effect (lower coverage) can be seen on no site preparation and pile and burn sites for ferns. The most common fern species found in this study was western brackenfern (*Pteridium aquilinum* (L.) Kuhn), which has been found to occur most frequently on open, disturbed sites [18]. Grasses also prefer open sites (thus the low coverage on the no site preparation sites). The small, disturbed burn piles may not be large enough openings to favor brackenfern establishment but large enough to encourage greater grass growth than in the non-treated areas. The mastication treatment had the lowest combined percentages of forbs, ferns, and grass but the highest percent substrate cover of solid wood, which may limit the regeneration of these smaller lifeforms for years after the treatment (Figure 2a).

### 4.2. Regeneration and Growth

It has been assumed that more light and growing space is needed to regenerate shade-intolerant species such as western larch and, to a certain extent, western white pine. However, for the six tree species we evaluated, sites do not need to be in full sun for regeneration establishment. An opening size between 55% and 92% favored a diversity of tree species (Figure 4). Sites that had opening sizes greater than 92% canopy opening also favor western larch, western white pine, and Douglas-fir which tend to have low to moderate shade tolerance. Grand fir and western hemlock, because of their more shade tolerance, had better regeneration establishment success in smaller openings. Although very shade tolerant, western redcedar did not regenerate in more closed conditions, which was an anomaly. Possibly, its inability to establish was more related to competition with grand fir and western hemlock. Jain and Graham (2005) stated that northern Rocky Mountain moist forests are disturbance dependent, with wildfires, insects, wind, and snow events always leaving standing dead trees that provide some cover for regeneration establishment. Thus, all native tree species, in order to gain a competitive advantage, have the ability to regenerate under some shade. This phenomenon was recognized by Haig et al. (1941) when they recommended that regeneration establishment is the most successful under some shade. Similarly, several authors have noted that a variable density shelterwood tends to favor the establishment of a wide range of tree species [19–21]. Jain et al. (2004) noted that a basal area that ranged from 10 to 25 $m^2$ $ha^{-1}$ produced canopy openings between 55% and 92%.

However, early growth is very different from regeneration establishment, making the opening size much more critical to maintain a competitive advantage. It is not surprising that as opening size increased so did 5-year height growth (Figure 5). However, we did not expect the prescribed fire to favor shorter trees when compared to the other site preparation methods. This was particularly true for grand fir. We hypothesize that water availability may have affected the growth for a few reasons: (1) the prescribed fire sites tended to be applied in larger openings (fewer residual trees to kill from the fire), which creates higher surface temperatures, (2) increasing evapotranspiration, and (3) soil drying resulting in different physiological responses of trees to drought. For example, Moran et al. (2017) [22] summarized several studies that noted that populations adapted to extended drought shift resource use from growth to maintenance, thus decreasing height growth. For Douglas-fir, vapor pressure deficit from stomatal closure during moisture stress reduces $CO_2$ uptake [23] affecting growth potential. In contrast, the other site preparation treatments, such as mastication, insulate the soil with organic matter, maintaining higher soil moistures [11], and these treatments also tended to be placed in areas with smaller opening sizes, offering more protection to established trees during the heat of the day.

### 4.3. Vegetation Diversity

The free selection silviculture system was designed to enhance vegetation diversity. Our silvicultural treatments resulted in abundant regeneration of six different tree species (Figures 3 and 4) and a diversity of lifeforms. We determined that canopy opening tends to influence the abundance of lifeforms and tree species particularly in canopy openings of 55% to 92% and on pile and burn site preparations (Figure 6). However, with such an abundance of vegetation, stand tending

will be required to ensure that the preferred species are given the competitive advantage. Species such as western larch do not compete well in high density plantations, thus they will need to be released so they can continue to grow to maturity. Not only are trees of value but also other lifeforms such as shrubs necessary for sustaining wildlife habitat. We noted that many of the shrubs had been browsed indicating that ungulates such as elk, deer, moose, and other browsing species were feeding in the area.

*4.4. Limitations and Future Research*

This study is somewhat limited since it was located only at Priest River Experimental Forest. Additional sites would provide the opportunity to validate the results. There is an opportunity to conduct validation on a similar study located on the Deception Creek Experimental Forest (also in northern Idaho but south of PREF) and there are opportunities to monitor similar sites that are being implemented by forest managers throughout the Inland Northwest (Idaho, eastern Washington, and western Montana). This study was not a completely crossed factorial design because it is not possible to implement a prescribed fire in places that are dominated by non-fire-resistant species, such as western redcedar; therefore, prescribed fire only occurred in the larger openings (>70% canopy opening; see Table A1). We focused on the regeneration phase of the silvicultural system in this study. Future studies need to focus on tending treatments, such as cleanings and weedings, since our results showed that some treatments (especially pile and burn) favored the establishment of too many trees. These overstocked stands will require some tending for optimum stand development.

## 5. Conclusions

The free selection silvicultural system that we designed and implemented created a diversity of opening sizes with different levels of green tree retention (Table 2). All the canopy opening sizes had a diversity of lifeforms and tree species. Shrub cover appeared to be the most sensitive to site preparation and opening size, with cover higher in the larger openings and occurring primarily on masticated and prescribed fire sites. Grand fir and western white pine responded similarly after establishment; sites with prescribed fire tended to diminish grand fir growth and did not statistically influence western white pine growth.

Regeneration establishment is the most important step within a silvicultural system. This study illustrated that canopy openings from 55% to 92% and the pile and burn site preparation method favored the establishment of all six species (Figures 3 and 4). This diverse species composition provides future adaptive management opportunities. Tending methods can be developed to favor cover for wildlife habitat or favor disturbance resilient species to a range of disturbances. There are very few studies that focus on regeneration establishment in moist mixed conifer forests; yet this forest type dominates northwestern USA and southwestern Canada. Our results are not only relevant for informing the free selection silvicultural system but also inform any regeneration method focused on creating gaps and small openings in mixed conifer forests using multi-aged management strategies, such as those described by O'Hara (2014).

**Author Contributions:** Theory conceptualization, R.T.G. and T.B.J.; treatment implementation, T.B.J. and USDA Forest Service, Priest Lake Ranger District personnel; methodology and validation, T.B.J., J.C.B. and B.C.B.; formal analysis, T.B.J. and J.C.B.; resources, Rocky Mountain Research Station; data curation, T.B.J. and J.C.B.; writing—original draft preparation, T.B.J. and J.C.B.; writing—review and editing, T.B.J., J.C.B. and B.C.B. All authors have read and agreed to the published version of the manuscript.

**Funding:** This research was funded by the USDA Forest Service, Rocky Mountain Research Station.

**Acknowledgments:** We would like to thank the following for assistance in field measurements: Jonathan Sandquist, Michael Skandalis, Patrick Fekety, Robert Denner, and the students from the Whitefish High School GIS class and FREEFLOW Club (Whitefish, MT, USA). We would like to thank Lane Eskew for editing this manuscript. The three reviews we received from Forests improved the manuscript and we want to thank the reviewers for investing the time to provide comments.

**Conflicts of Interest:** The authors declare no conflict of interest.

**Appendix A**

This appendix includes the free selection concepts we applied in 2007, the experimental design (Table A1) and examples of different opening size (Figure A1). We returned in 2015 to quantify the outcome after regeneration establishment, growth, and understory vegetation response (tree age was 8 years old).

*Implementing Free Selection Concepts*

When we established the treatments in 2007, the free selection system by Graham and Jain (2005) [24] identified several questions to guide the preparation of the silvicultural prescription:

(1)   What is the management objective? We implemented the study on an Experimental Forest, thus the objectives were to create a diversity of forest conditions to enhance current and future research opportunities, to create a diversity of surface and crown fuels to alter fire behavior and effects, and to develop, implement, and evaluate an alternative silvicultural system designed to introduce diversity in forest composition and structure.

(2)   What is the theme or broad management goals? Goals included producing snags over time, increasing vegetation diversity thus diversifying wildlife habitat, favoring the abundance of disease-resistant species while still maintaining many tree species for adaption opportunities.

(3)   What are the desired future conditions? Using a range of canopy opening sizes and soil disturbances, we wanted to develop a landscape mosaic of different compositions and structures. Tree composition, depending on the growing space, would vary from sites containing fast growing western larch, western white pine, and ponderosa pine, and in moderate growing environments, western white pine, grand fir, and Douglas-fir. In addition, across the range of moderate growing conditions, we recognized that there would be environments that would favor western redcedar and western hemlock. Forest structures would range from shrub to stem initiation to stem exclusion. There would be continuous recruitment of snags over the entire landscape, ranging anywhere from 2 to 5 snags per hectare resulting from a diversity of disturbances (drought, insects, disease, wind, and snow).

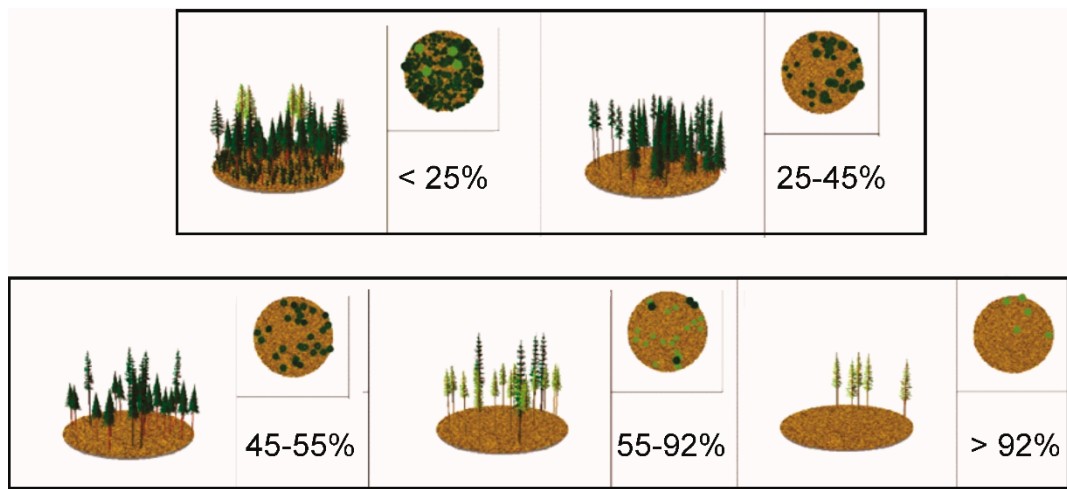

**Figure A1.** Examples of different canopy openings (%) using the Stand Visualization System [25].

**Table A1.** Regeneration harvest and site preparation study design [10]. Western white pine thresholds for regeneration establishment (25 to 55% canopy opening, competitive advantage [55 to 92%], and free-to-grow [>92%]) [9], soil substrates (blackened soil, mineral, organic, and masticated), and a combination of strip cuts, gaps, and circles were used to create the range of opening sizes. Site preparation treatments were replicated a minimum of five times each across the entire study area and included grapple pile and burn (pile and burn), shredded wood (mastication), prescribed fire, and no site preparation (no treatment). The mechanical treatments site preparation treatments were randomly applied on slopes less than 50% because machine limitations prevented grapple pile machines and masticators to operate on steep slopes. The prescribed fire was randomly applied on steep slopes and when residual tree species were fire resistant.

| Unit Number | Regeneration Treatment | Replicate | Canopy Opening Target (%) | Site Preparation |
|---|---|---|---|---|
| | | Block 1 | | |
| 17_1 | 16 m strip | 1 | 55 | Pile and burn |
| 17_2 | 16 m strip | 2 | 55 | Pile and burn |
| 16c1 | 16 m strip | 3 | 55 | Pile and burn |
| 18 | 31 m strip | 1 | 70 | Prescribed fire |
| 10 | 31 m strip | 2 | 70 | Mastication |
| 14 | 31 m strip | 3 | 70 | Mastication |
| 20 | 47 m strip | 1 | 85 | Pile and burn |
| 12 | 47 m strip | 2 | 85 | Mastication |
| 21 | 47 m strip | 3 | 85 | Pile and burn |
| 19 | 62 m strip | 1 | 100 | Prescribed fire |
| 11a | 62 m strip | 2 | 100 | Prescribed fire |
| 13 | 62 m strip | 3 | 100 | Pile and burn |
| 8_1 | 0.4 ha circle | 1 | 92 | Mastication |
| 8_2 | 0.4 ha circle | 2 | 92 | Mastication |
| 8_3 | 0.4 ha circle | 3 | 92 | Mastication |
| 23 | <0.2 ha gap | 1 | 25-40 | Mastication |
| 15 | <0.2 ha gap | 2 | 25-40 | Mastication |
| 9a | <0.2 ha gap | 3 | 25-40 | Mastication |
| 9b | <0.2 ha gap | 4 | 25-40 | Prescribed fire |
| 9c | <0.2 ha gap | 5 | 25-40 | Mastication |
| 22 | <0.2 ha gap | 6 | 25-30 | Pile and burn |
| | | Block 2 | | |
| 16a | 16 m strip | 1 | 55 | Mastication |
| 16b | 16 m strip | 2 | 55 | No treatment |
| 16c | 16 m strip | 3 | 55 | Pile and burn |
| 5_1 | 31 m strip | 1 | 70 | Prescribed fire |
| 7a | 31 m strip | 2 | 70 | Mastication |
| 7b1 | 31 m strip | 3 | 70 | Mastication |
| 7b2 | 47 m strip | 1 | 85 | Pile and burn |
| 6_1 | 47 m strip | 2 | 85 | Prescribed fire |
| 6_2 | 47 m strip | 3 | 85 | Prescribed fire |
| 5_2 | 62 m strip | 1 | 100 | Mastication |
| 4b | 62 m strip | 2 | 100 | Mastication |
| 7c | 62 m strip | 3 | 100 | Prescribed fire |
| 4a | 0.4 ha circle | 1 | 92 | Mastication |
| 8_4 | 0.4 ha circle | 2 | 92 | Mastication |
| 8_5 | 0.4 ha circle | 3 | 92 | Mastication |
| 1 | <0.2 ha gap | 1 | 25–40 | Pile and burn |
| 2a | <0.2 ha gap | 2 | 25–40 | No treatment |
| 2b | <0.2 ha gap | 3 | 25–40 | No treatment |
| 3 | <0.2 ha gap | 4 | 25–40 | Prescribed fire |

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
