# Peer review of "Canopy Opening and Site Preparation Effects on Conifer and Understory Establishment and Growth after an Uneven-Aged Free Selection Regeneration Harvest in the Northern Rocky Mountains, USA"

_forests, doi:10.3390/f11060622_

Round 1

Reviewer 1 Report

REVIEW forests-806005. Canopy Opening and Site Preparation Effects on Conifer and Understory Establishment and Growth After an Uneven-Aged Free Selection Regeneration Harvest

13/05/2020.

This study reports the results of applying a free selection treatment in a mixed-conifer forest of Western USA in 2007. The manuscript reports tree regeneration establishment, tree growth, and diversity of vegetative lifeforms (trees, shrubs, forbs, ferns, and grasses), 8 years after treatments were applied, as a function of opening size and site preparation. Several strata were considered according to canopy opening classes and site preparation and a total of 65 plots were assigned to the strata in an unbalanced design. As a result of the process, authors obtained very relevant silvicultural information in the form of parameters to design the regeneration and early growth of several species with different degrees of shade tolerance. The study provides scientific information in the field of silviculture aiming to creating multi-aged forest structures and diverse vegetative compositions in moist mixed-conifer forests of North America. The results can be of interest for temperate forest in other parts of the World. The manuscript is well written, and the contents fits well in the aims and scope of the journal.

I enjoyed very much reading this “pure silviculture” manuscript in these dark times. Thanks for the opportunity of reviewing.

I’m providing below some comments that would hopefully be relevant to improve the manuscript, particularly the way of showing the Results.

L43. The study refers to North America exclusively without mentioning this specifically. Note for example that mimic forest conditions that occurred at some point in the previous two centuries in the Old Continent means going back only to Napoleonic Wars, when natural forests were already quite battered by centuries of wars…

L46-48. The North American context is also evident here. Please, refer to North America in the text. Maybe also in the Title.

Section 2.3. Canopy cover classes. I’m confused about the actual number of classes considered. 4 classes are mentioned in L159, but 5 in L 162 (making 20 the total possible number of strata, 5x4), 4 are again considered in table 1, but 20 sampling strata in L173 (instead of the 16 the reader are expecting). Moreover, in the result section, 5 classes are definitively considered, but with different ranges (<25, 25-45, 45-55,55-92 or >92%) than the ones proposed in M&M (0-25, 25-50, 50-75 and >75%).

L197. 0.04 instead of 0/04

L222. Not sure what the expected value stands for. It probably refers more to densities of trees than to frequencies (as frequencies are the number of times a data value occurs). I consider this important for the reader to correctly interpret Figures 3 and 4. If I’m right, Figure 3a means for example that regeneration density of Larix occidentalis is more than expected for the pile&burn treatment. Also, the sum of all the deviations for the four treatments should be 0.

L231. Results. Some data on basis dendrometric parameters of the standing trees (the large ones not felled in the free selection treatment) would be needed. Otherwise, any information on the variables measured as explained on L196-204 would be available in the ms.

Results. What about Pinus ponderosa? No information recorded for this species?

Figures 3&4. Another problem with the chi-gram way of showing the results is that the reader ends up without any clue of the regeneration density found for each species, in terms of plants per ha. Please, add this information.

Figure 2. Please, add error bars for Figure 2.

L250. Not clear to me, although I’m not a native English speaker. Please, consider this alternative: In addition, western white pine and grand fir successfully established on the mastication site preparation, whereas western larch successfully established on the prescribed fire site preparation.

Results on regeneration and canopy opening classes are apparently very relevant but are only incompletely shown. For example, the Pinus monticola regeneration facilitation effect mentioned in L264 is not what the reader can expect from Figure 4d, as deviations are negative from what is expected.

In fact, the chi-gram approach does not provide much information and some of the sentences mentioned in the text (particularly the threshold of canopy opening for shade-tolerant species establishment) are not possible to visualize.

I suggest here the incorporation of additional figures, maybe plots of actual canopy opening values (as a continuous variable) against the regeneration densities for the different species.

As regards the growth data, I also consider the results are incompletely shown. Regressions are provided for two species, but descriptive statistics would be most welcome for all six conifers.

Please, add error bars for Figure 6.

L347. Better keep the same order of shade tolerance: very little (Larix occidentalis) to moderate (Pseudotsuga menziesii & Pinus monticola) shade tolerance.

L396. Conclusion. I recommend re-writing completely this section, keeping some of the ideas for the discussion and placing here concisely the Conclusions derived from the study.

Figure A1. This is a very relevant set of figures obtained apparently from the SVS. I have a number of suggestions for these figures: a. please, cite the Stand Visualization System as source of the figure. b. Figures are very small and with low resolution, I think they could be used as graphical ways of showing the results and discussion. Each species should be identified. C. Please, note the canopy opening classes are reversed (i.e., the right side refers to canopy opening 0-25%). D. Please, note also the profile views are not representing all the trees in the plots.

Reviewer 2 Report

Please see my comments in the pdf-file. Most important is to improve the Results (and Appendix) -figures, and to improve the title, objectives statement, and M+M. 

I look forward to reading the improved, published version of this article!

Reviewer 3 Report

Review of: Canopy Opening and Site Preparation Effects on Conifer and Understory Establishment and Growth After an Uneven-Aged Free Selection Regeneration Harvest

For: Forests

May 2020

 Summary:

The researchers compared tree growth, substrate condition, and understory vegetation among four classes of canopy opening and four types of site preparation treatments following canopy harvest.  Treatments that included burning exposed more soil and resulted in less solid and rotten wood than other treatments.  Pile and burn produced the most tree regeneration, as did canopy opening of at least 55%. 

Strengths:

This manuscript is well written, the study is large in scope, and the results will be useful for forest management in the western United States.

Weaknesses:

The graphs are inconsistent in organization and there is not a consideration of biodiversity among the treatments.

Specific Suggestions:

  1. On line 26, replace “were” with “was”.
  2. In the abstract, the Results section does not mention the understory results.
  3. On line 41, replace “are” with “is”.
  4. On line 134, “increasing vegetation diversity” is described as a goal of the effort, yet there is not a consideration of biodiversity in the study. Could the researchers use the data that they have to assess differences in biodiversity among the treatments?
  5. In the Experimental Design, it is not clear how the study sites were assigned to the different site preparation treatments. Could the authors provide a brief statement of how these assignments were made?
  6. On lines 167 and 171, all uses of “stratum” and “stratums” should be replaced with “strata”.
  7. On line 180, the close parentheses after “[1.0 ft]” should be deleted. Also for consistency, the brackets around “1.0 ft” should be parentheses.
  8. In Statistical Analysis, there is not a description of how the understory data were analyzed. I am guessing the same approach was used for these data as was done for the tree species. 
  9. The organization of Figs. 2 and 6 is different from that for Figs. 3 and 4. In the former, the panes shows differences among response variables within a treatment, but in the latter, the panes show differences among treatments within a species.  It would make more sense to keep a consistent format, or even make these grouped bar graphs (grouped by treatment).
